# Machine Learning Strategy for Soil Nutrients Prediction Using Spectroscopic Method

**DOI:** 10.3390/s21124208

**Published:** 2021-06-19

**Authors:** Janez Trontelj ml., Olga Chambers

**Affiliations:** Faculty of Electrical Engineering, Trzaska cesta 25, 1000 Ljubljana, Slovenia; janez.trontelj2@guest.arnes.si

**Keywords:** machine learning, nutrients prediction, soil spectra, soil analysis, soil category, precision farming

## Abstract

The research presented in this paper is based on the hypothesis that the machine learning approach improves the accuracy of soil properties prediction. The correlations obtained in this research are important for understanding the overall strategy for soil properties prediction using optical spectroscopy sensors. Several research results have been stated and investigated. A comparison is made between six commonly used techniques: Random Forest, Decision Tree, Naïve Bayes, Support Vector Machine, Least-Square Support Vector Machine and Artificial Neural Network, showing that the best prediction accuracy cannot always be achieved by the most common and complicated method. The influence of the chosen category for nutrient characterization was investigated, indicating better prediction when a multi-component strategy was used. In contrast, the prediction of single-component soil properties was less accurate. In addition, the influence of category levels was not as significant as expected when choosing between 3-level, 5-level or 13-level nutrient characterization for some nutrients, which can be used for a more precise nutrient characterization strategy. A comparative analysis was performed between soil from a local farm with similar texture and soils collected from different locations in Slovenia, which gave a better prediction for a local farm. Finally, the influence of principal component analysis was validated using 5, 10, 20 and 50 first principal components, indicating the better performance of machine learning when using the 50 principal components.

## 1. Introduction

Artificial intelligence is the most rapidly growing area integrated into almost all aspects of human life. It has been proven to be a helpful tool that provides a second opinion, highlights poorly visible information, and predicts behavior based on previous experience and learning algorithms [1,2,3,4,5,6,7,8]. Usually, the results rely on various factors such as research dataset size, the parameters of the algorithm, the soil type and the categories to be estimated. Due to the considerable variation in the factors and their combinations, it is almost impossible to replicate the published results with the same accuracy when different research datasets are used. Nevertheless, the published and our observations can give a research direction to develop an advanced strategy for predicting soil properties. This research investigates the influence of the common machine learning techniques and other parameters affecting its performance to create a suitable strategy for accurate soil characterization using the Ultraviolet-Visible Spectroscopic (UV-VIS) method.

The first research focus is to determine an appropriate machine learning approach for the most accurate soil properties prediction. Some researchers prefer complex methods when others declare good results with a simple technique such as the Partial Least Squares Regression (PLSR) method [9,10]. In [7], the authors showed that the PLSR, based on UV-VIS and VIS -NIR spectra without selection of spectral variable selection, provided the ability to distinguish between high and low values for Nitrogen (N), Organic Carbon (OC), Magnesium (Mg) and other components. However, studies [5,11,12] indicate that the relationship is not always linear, so PLSR could be considered inadequate for modeling soil properties. In [11], the authors found consistently higher performance of machine learning methods over simpler approaches in spectroscopy. Helong Yu et al. in [13] reported the decline in the use of some models such as Support Vector Machines (SVM) and multivariate adaptive regression spline, giving way to more advanced alternatives such as Random Forest (RF). Other research showed higher effectiveness of Neural Networks (NN) [10,14,15], Decision Tree (DT) [16], Naive Bayes [17,18], etc. Recently, Convolutional Neural Networks (CNNs) have received much attention in object classification [19]. Therefore, it is unclear if there is a correlation between the methods’ complexity and prediction accuracy. It is also important to understand the methods’ advantages and disadvantages to provide a correlation between the obtained results and the methods complexity. Six common techniques RF, DT, Naïve Bayes (NB), SVM, Least-Square Support Vector Machine (LS-SVM) and Artificial Neural Network (ANN), having a different degree of complexity, were used for comparative analysis.

The next research challenge is to investigate the Principal Components (PCs) influence on the overall performance. To capture the maximum variation in spectral data, many researchers combine the PCs with a classification model, which increases feature discrimination and improves computational efficiency. The principal component analysis is an essential part of machine learning that can significantly improve classification accuracy and time efficiency. However, researchers still need to be very precise when defining an appropriate number of components for individual dataset and measuring method. In this research, we compared results corresponding to the first 5, 10, 20 and 50 principal components on the prediction accuracy to investigate their influence on the overall analysis.

Another research point is to determine an appropriate soil category for machine learning without significant accuracy loss. Researchers mainly focused on a single-component problem, which usually provides good prediction accuracy. Calibrations for N and OC are probably the most common when the spectroscopic method is used. According to [8], OC and total N are those with the best chance of success. However, soil is a complex mixture of water, stones, micro and macro nutrients, which increase the solution complexity and decrease the prediction accuracy. For example, Padarian J. in [19] reports better performance of machine learning for the multi-tasking model where they predicted multiple soil properties. Ruder S. states that the multi-tasking model reduces the risk of overfitting [20]. Therefore, a single-component and multi-component strategies were compared in this paper to distinguish the difference between their performance. The second part of this problem is also a definition of the number of nutrients levels. The selection of nutrient categories is generally based on the requirements of agriculture, the yield level, and the type and nature of the soil in question [21,22]. When selecting a category for analysis, it is vital to consider the overall crop requirements and the appropriate accuracy rate achieved by the chosen machine learning algorithm. Intuitively, it is expected that the prediction accuracy for a category with two levels will be more accurate than with five levels. Researchers mainly divide fertility levels into three main categories: below optimum, optimum, and above optimum [23,24,25,26,27,28], where the “low” or “high” category can increase or decrease the fertilizer recommendation by 25% or 30% of a general recommendation. Heckman J. R. et al. [25] further divided into subcategories: very low, low and medium. In Johnstown Castle, soil analysis levels are classified into four categories: define, likely, tenuous and none. In some states, soil characterization is done with ratings of very low, low, medium, high, very high. Other six-level soil nutrients are grading i.e., very low, low, medium, moderately high, high, and very high [29,30]. There is no more than six-level grading reported in the literature for spectroscopic methods. In our research, three categories having 3, 5 and 13 levels were used to validate the category influence on the prediction performance using a comparison of the machine learning results, where 13 levels were selected to have a representative contrast to the other two. To the best of our knowledge, no such analysis was reported in the literature.

Finally, the influence of the soil location on the prediction accuracy was investigated. We found out the importance of the database representativeness for the area under analysis with minimum soil variability [31,32]. Intuitively, it is clear that a more extensive data set will allow for more accurate soil characterization. However, there is no clear rule on how large a dataset needs to be. The general trend is that the prediction accuracy increases with the number of samples [33]. In [19], the authors observed that it varies greatly depending on the machine learning method. The commercially available database AgroCares nutrient scanner [23] is now used in 22 countries and provides only brief soil characterization. Benedet L. et al. in [34] suggested that more locally specific models can produce better prediction accuracy depending on the target soil property. They showed good coefficients of determination for soil components because their study area contained only two soil classes and thus less soil variability. In order to investigate the influence of the dataset on the prediction accuracy, two datasets were created, where one contains soil from a local area and another one contains soil collected from all over Slovenia, having different chemical and textural properties. Due to the project requirements, the only three nutrients were selected for analysis: Phosphorus (P), Potassium (K) and Magnesium (M) as they are the most common for local agriculture in vineyards and orchards.

The paper is organized as follows. Section 2 provides information about standard machine learning techniques used for soil analysis. Section 3 describes our research dataset and procedure for spectroscopic data acquisition. Results of the comparative analysis are reported in Section 4, providing important information for soil characterization strategy development. Discussion about the obtained results is in Section 5, followed by a Conclusion.

## 2. Analysis of Machine Learning Methods

The analysis presented here is not exhaustive but considers almost 50 relevant publications from scientific journals, mainly dealing with soil spectra measurement in the UltraViolet-Visible Near-Infrared (UV-VIS-NIR) range [11,14,35,36,37,38,39].

Commonly, soil analysis includes Binary Tree (BT), Support Vector Machine (SVM), Naïve Bayes (NB), Artificial Neural Network (ANN), Cubist regression (CB), Principal Component Regression (PCR), Partial Least Square Regression (PLSR), Least-Square SVM (LS-SVM), Extreme Learning Machines (ELM), Ordinary Least Square Estimation (OLSE), Ant Colony Optimization-interval Partial Least Squares (ACO-iPLS), Deep Learning (DL), Fully Connected Neural Network (FNN), Multiple Linear Regression (MLR), Regression Tree (RT), Random Forest (RF), Generalized Additive Model (GAM), Convolutional Neural Network (CNN).

In Table 1, a comparison of some relevant machine learning methods is presented.

Based on our experience, machine learning’s most common advantages and disadvantages are summarized in Table 2 and Table 3.

## 3. Materials and Methods

This section describes research datasets of soil samples and the measuring procedure used in our analysis. Using the obtained measurements of corresponding research datasets, the results of comparative analysis for different ML techniques and parameters affecting their performance, such as principal components, soil structure variation and category type for nutrients description, are reported. The spectroscopic measurements were performed under laboratory conditions keeping constant temperature and humidity during all research studies. MATLAB software [48] was used to implement a machine-learning algorithm to classify and predict the soil category.

### 3.1. Research Dataset

Soil samples for laboratory analysis were collected at a depth of 0–30 cm from different locations in Slovenia, naturally dried, grounded and passed through a 2-mm sieve. Two research datasets were created having entirely different properties. The first dataset consists of 50 soil samples, which is called here as the Global soil dataset. Soils for this dataset were collected in different locations having a large texture and chemical content variation.

The second dataset consists of 8 soil samples collected at a local farm that has been fertilized, which is called here as Local soil dataset. The research datasets are relatively small due to the wide variety of chemical combinations among the soil samples. Therefore, seven subsamples of each soil were randomly collected and measured. The final dataset corresponding Global soil dataset consists of 350 samples, and the dataset corresponding Local soil dataset consists of 56 samples.

In order to perform machine learning validation, the dataset of sub-subsamples corresponding Global soil dataset was split into test and training sets. Four sub-samples corresponding to the same soil were used to create the training set, and three sub-samples were used to create the test set. This is a standard procedure when soils with known spectra and chemical properties are included in the training set to validate the method’s selectivity. Such a strategy allows the prediction of soil properties with higher accuracy.

Chemical characterization of the samples was performed in a certified laboratory in Slovenia, i.e., Agriculture Institute of Slovenia [49], to perform accurate machine learning and validation of the results.

### 3.2. Spectroscopic Data Acquisition

Data acquisition of the spectra was performed within the UV-VIS range, i.e., (200–11,000) nm, and joined into research datasets of measurements corresponding to Global and Local soil datasets.

A deuterium halogen light box was used as the light source. The light reflectance from the sample was measured by placing 5 g of air-dried sieved sample into a quartz glass petri dish three mm-diameter, as shown in Figure 1. The set-up includes a fiber-coupled spectrometer FCR-7UV200-2-ME from Avantes that is fixed perpendicularly to have a 3 cm distance between the probe and the samples. The light from a light source is sent through six illumination fibers to the sample, and the reflection is measured by a seventh fiber in the center of the reflection probe tip. The AvaSpec-ULS2048CL-EVO-RS and AvaSpec-HSC-TEC perform the light measurement in the UV-VIS-NIR range of the electromagnetic spectrum. Spectra normalization was performed by dividing soil reflectance spectra by the white body reflectance spectra used here as a reference.

## 4. Research Results

Initially, the influence of the category of the nutrients levels was studied. Three different categories for soil characterization were defined in this research using 3-levels (Category III), 5-levels (Category II) and 13-levels (Category I) (see Table 4). For example, if soil contains 5 mg of the phosphorus, 15 mg of potassium and 35 mg of magnesium, the soil class would be “2-5-11” for Category I, “1-2-4” for Category II and “1-1-3” for Category III, where the first position corresponds with phosphorus, the second with potassium and the third position is for magnesium.

Using the grading system proposed in Table 4, the comparison between soil properties’ predictions was performed and presented in Table 5. As expected, the prediction accuracy decreases with increasing gradation. Nevertheless, the difference in accuracy between neighbors’ categories is not significant for some nutrients that can be used for more accurate nutrient content characterization. Thus, a more accurate prediction of phosphorus and potassium can be achieved without a significant loss of accuracy.

Typically, a classification task involves the prediction with a single-component class label. Alternatively, it may involve the prediction with multi-component class labels. This implies that class membership is not mutually exclusive. Figure 2 describes the formulation of the class label corresponding to the ground data from the training set. The left column corresponds to the single-component class label formulation, where only information about one nutrient is used for classification. In this case, machine learning assumes that the input belongs to only one class and does not depend on other components’ variations. The final class label consists of values corresponding to the selected category of nutrient values shown in Table 5.

Table 6 shows the results obtained for soil nutrient prediction when a single-component and multi-component class label definition was used. It can be seen that a multi-component strategy is more effective than a single-component prediction. The more soil components such as texture, pH, organic mass, micronutrients, etc., are involved in the classification, the better the soil properties can be predicted. This behavior is similar to that described in [19], where the accuracy increased continuously with the number of tasks.

Six common ML methods were selected to evaluate their influence on soil property prediction. The best parameters were adjusted and optimized for each method. To evaluate the accuracy of the performance of ML, the precision metric was used, which indicates how many of the data were predicted correctly. Figure 3 and Figure 4 show the results of predicting soil properties according to the selected ML method using category system II defined in Table 5 and defining multi-component classes for global and local research datasets, respectively. It can be concluded that ANN and LS -SVM outperformed the other methods and provided better prediction for selected nutrients, while NB showed less effective performance. The table shows lower sensitivity of the UV-VIS range for the presence of P, where magnesium was easier to detect with all machine learning techniques. The better performance of the machine learning techniques can be observed for the local dataset. This is consistent with observations in the literature and can be explained by a more homogeneous texture of the local dataset and smaller intra-class variation of the unknown components. This is an important observation when the research project is limited by budget.

Using the machine learning settings defined in the previous experiment, the influence of principal components was investigated. The first 5, 10, 20 and 50 principal components with the highest score were selected for analysis compared with the result obtained without dimensionality reduction. The obtained results are presented in Figure 5 and show the improved accuracy of soil category prediction for all methods except ANN, where the result was improved only for magnesium. It can be noted that the variation in results is similar for each nutrient. The influence of PC can be observed to varying degrees for each method’s accuracy. It can be found that SVM performance varies less and therefore is more stable when the performance of NB, DT and RF strongly depends on the selected PC. In general, the results of ML techniques show good performance for PC20.

## 5. Discussion

In this section, we discuss the results and observations used in developing the machine learning strategy for soil nutrients prediction with the spectroscopic method.

The obtained results reported in the previous section indicate ML methods’ advantages and disadvantages, allowing creating an optimal strategy for research. It was shown that the method’s high level of performance complexity does not always guarantee a satisfying level of performance accuracy. The LS-SVM and ANN showed the most promising results. Nevertheless, they do not show the best result for a Local soil dataset in this research. For example, the RF showed better performance for a potassium prediction and DT showed similarly as ANN result for magnesium prediction. Results corresponding Global soil dataset indicates that LS-SVM overperforms ANN for phosphorus and magnesium prediction. Moreover, the computation time for processing ANN was always the highest, taking up to a few minutes to compute on a standard personal computer. It is worth mentioning that every time the category is changed, the machine learning parameters have to be changed as well. For example, for our research dataset, ANN gave the best results when 10 layers were used for categories with 5 levels, while for categories with 13 levels, the best result was obtained with 20 layers.

The correlation between the class definition and the output results for soil property prediction was investigated to estimate the model’s sensitivity to the category. The results show an increase in prediction accuracy with decreasing categories. It was observed, that miss-classification is mainly happening for data having category properties similar to the neighbor category because of the small difference between corresponding measurements. This may explain, why the increase of the categories leads to the increase of the miss-classifications. Nevertheless, a slight difference was observed between the results corresponding to different categories for some nutrients. This can be used if a more accurate soil characterization needs to be performed. In our research, we used near evenly distributed categories. However, it must also be taken into account that the category levels may be unevenly distributed. We believe that several categories may be needed not only to characterize the nutrients in the soil but also to characterize measuring technique sensitivity and selectiveness. In this research, 13-class category characterization was used based on our project requirements.

The results reported in Table 6 show the importance of multi-component analysis, also known as multi-task learning. The prediction of soil properties has better accuracy when all parameters are used during the learning process compared to the prediction performed when only the single parameter is learned. This difference is especially critical when more nutrients levels are used (i.e., Category I in Table 4). The strategy of selecting multi-component analysis reduces the risk of overfitting and thus the overall accuracy increase.

Furthermore, it was shown that both, soil structure and soil properties, significantly affect the performance of the ML methods. Consequently, a better accuracy can be achieved within a local farm due to the homogeneous texture and micronutrients variation.

## 6. Conclusions

The best results in our research can be obtained for a local farm using ANN. For a Global soil dataset, the best strategy would be using LS-SVM with 50 PCs and multi-component categories for data labelling. Category III provided the best result. Nevertheless, an acceptable level of accuracy can be achieved for potassium and magnesium when using Category II for classification.

This study was performed only for phosphorus, potassium and magnesium. Nevertheless, similar analyses can be repeated for other soil components such as nitrogen and organic mass. We believe that the presented results can clarify the most common parameters that affect machine learning performance and help to select the optimal strategy for a particular research task in soil study.

The influence of the machine learning techniques and their parameters was investigated and discussed in this paper. The performed comparative analysis indicates their different influence that may increase or decrease the overall soil analysis accuracy. The most critical observations for soil spectroscopy strategy development made during our research are listed below:Multi-component analysis outperforms single-component soil prediction;There is no universal ML technique that can be used with the same accuracy for different research datasets and problems. Nevertheless, the comparative analysis shows a good performance of LS-SVM and ANN for our research dataset;The dataset of measurements corresponding to the local agricultural soil provide more effective prediction of soil properties than a dataset consisting of different geographical positions;The use of principal component analysis has been shown to improve overall prediction accuracy. By comparative analysis, it was estimated that 50 principal components are the optimal number for an accurate result;Repeated measurements followed by averaging of the same soil measurements can improve the accuracy of soil property characterization.

The presented results confirm our initial hypothesis that the Machine Learning significantly improves the accuracy of soil property prediction.

## Figures and Tables

**Figure 1 sensors-21-04208-f001:**
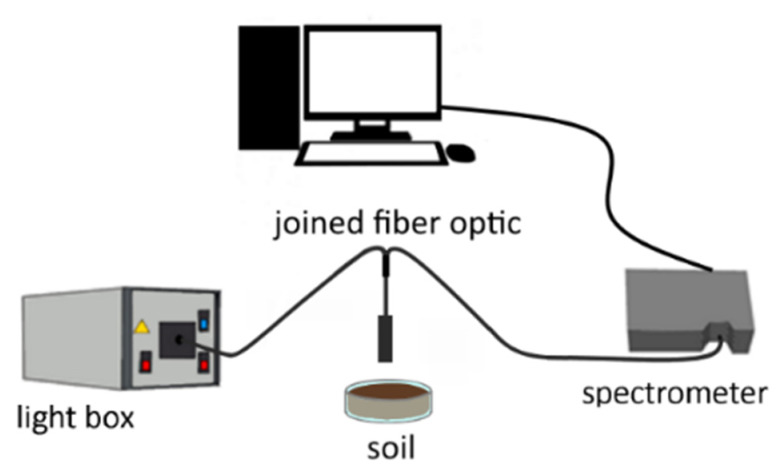
The experimental set-up for spectroscopic data acquisition.

**Figure 2 sensors-21-04208-f002:**
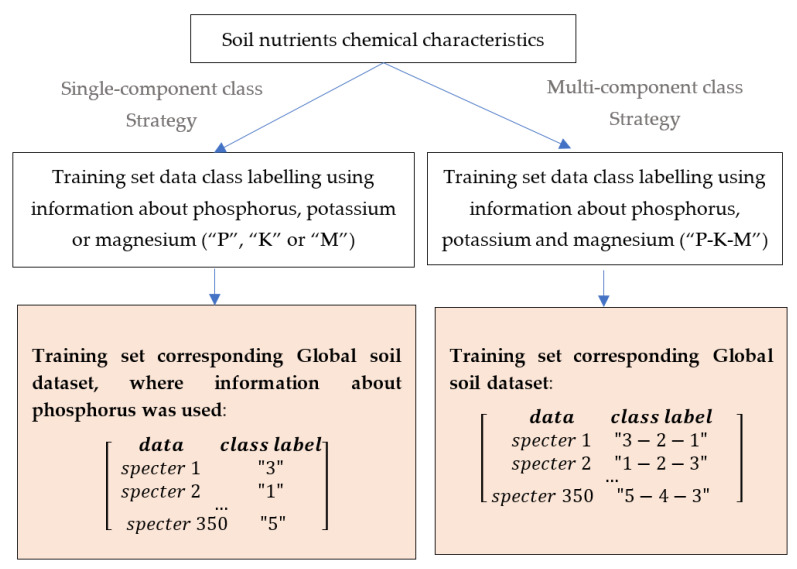
Flow diagram of the Training set class labelling using single-component (left) and multi-component (right) strategy, corresponding Category II.

**Figure 3 sensors-21-04208-f003:**
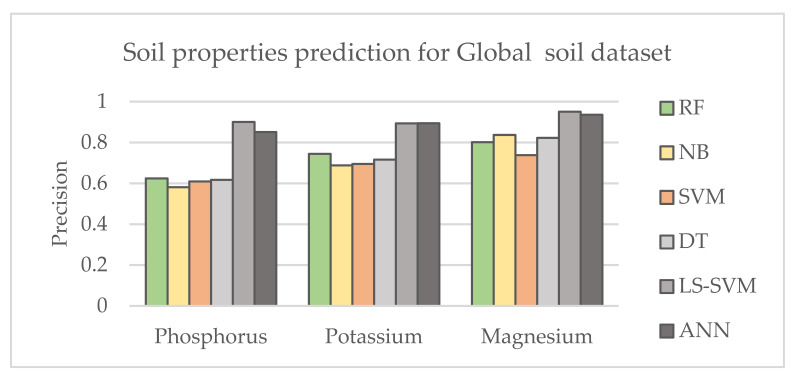
Comparison of the results corresponding to different machine learning techniques for the Global soil dataset using Category II system for category labelling.

**Figure 4 sensors-21-04208-f004:**
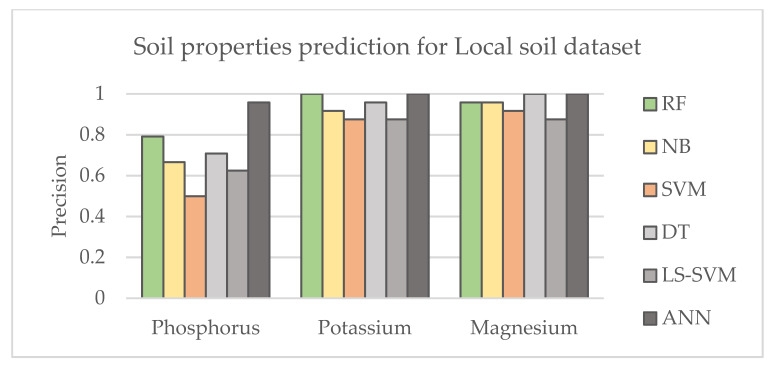
Comparison of the results corresponding to different machine learning techniques for Local soil dataset using Category II system for category labelling.

**Figure 5 sensors-21-04208-f005:**
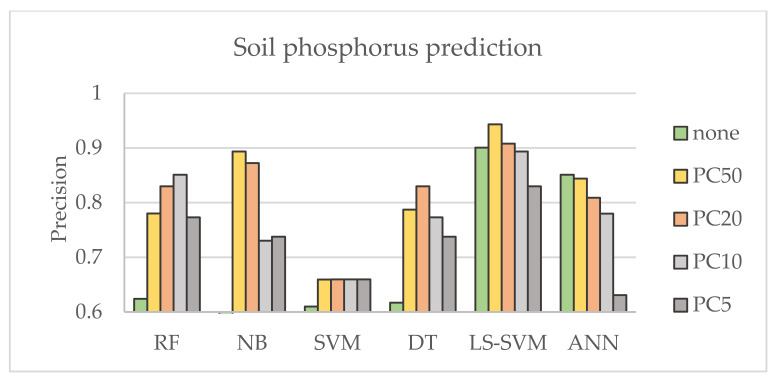
The precision corresponding to the Global soil dataset when a different amount of the principal components is used.

**Table 1 sensors-21-04208-t001:** Comparison of machine learning methods within VIS-NIR range.

Methods Used for Comparison	Best Method	**Reference**
PCR, PLSR, LS-SVM and CB	LS-SVM for OC, CB for N	[24]
PLSR, LS-SVM, ELM and CB	ELM for OC	[10]
ANN, RF PLSR and CB	CB for OC, PLSR for N	[29]
PLS, BPNN, ELM	ELM for OC and N	[30]
PLSR, BPNN and GA-BPNN	GA-BPNN for N, P, K	[40]
PLS and SVR	SVR for available K	[41]
LS-SVM and PLSR	LS-VM for N, P and K	[42]
OLSE, RF and ELM	ELM for N	[43]
AOC-iPLS, RF and RF-SVM	RF-SVM for OC	[44]
PLSR, SVM, RF, ANN and DL	ANN for Cr and Al	[45]
PCR, PLSR, LS-SVM, BP-NN	BPNN and LS-SVM for different nutrients	[46]
RF, Cubist, SVM, ANN and MLR.	RF for OC	[37]
13 ANN models	GRNN for nutrients	[47]

**Table 2 sensors-21-04208-t002:** Comparison of advantages of machine learning techniques.

Method	Advantages
Regression	Low computation time
	Performs well with large datasets
	Reduce data dimensionality
	Provide a feature selection
	Easy to implement
DT	Can be effectively applied for the nonlinear problem
	Performs well with large datasets
	In built feature selection procedure
	Easy to implement
SVM	Can be effectively applied for the nonlinear problem
	Performs well when data dimensionality is greater than the number of samples
	Low risk of the over-fitting
NB	Can be effectively applied for the nonlinear problem
	Low computation time
	Suitable for multi-class problems
	Effective for small training datasets
	Easy to implement
	Robust to small dataset changes
	Probabilistic predictions can be obtained
RF	Can be effectively applied for the nonlinear problem
	May be applicable to soils under a great variety of environments
	Act to reduce bias
	Performs well with large datasets
	Overfitting is less common
	Accommodate random inputs and random features
	Can be used for classification as well as for regression
NN	Can be effectively applied for the nonlinear problem
	Effective in many applications
	Defined fault tolerance that makes classification more robust
	Robust to small dataset changes
	Can perform in parallel without affecting the system

**Table 3 sensors-21-04208-t003:** Comparison of disadvantages of machine learning techniques.

Method	Disadvantages
Regression	Do not deal with nonlinear problems
	over-fitting may occur
DT	Over-fitting may occur
	Non-robust to small dataset changes
	Input parameters, such as nods numbers, need to be defined manually
SVM	Non-robust to small dataset changes
	Is not suitable for large datasets, where data dimensionality is smaller than number of samples
	Effective kernel function is not easy to define
	Large computational time for large datasets
	Different impact of the weights parameters that is not easy to visualize their impact
	Needs adaptation for multi-class problems
	Not easy to implement
NB	Assigning zero probability to a categorical variable is not available
	variable loss of accuracy
RF	Had difficulty predicting high and low laboratory measured values, underestimating and overestimating them, respectively
	Number of trees need to be defined manually
	Long computation time
	Large computational power is required due to the large amount of the trees created by the algorithm
NN	Effective architecture parameters need to be defined manually
	Difficult to implement
	Require large training dataset
	Large computational power is required
	Weights are assigned randomly, so as to acquire high accuracy, the process of training the data must be iterative
	Classes have to be translated into numerical values
	The duration of the network is unknown

**Table 4 sensors-21-04208-t004:** Category range specifications.

Score, mg/100 g	Category I	Category II	Category III
0–3	1	1	1
5–7	2		
8–10	3		
11–13	4	2	
14–17	5		
18–20	6		2
21–23	7	3	
24–27	8		
28–30	9		
31–33	10	4	3
34–37	11		
38–40	12		
>40	13	5	

**Table 5 sensors-21-04208-t005:** Prediction results corresponding to the Global soil dataset analysis for different categories using the DT method.

Categories	Phosphorus	Potassium	Magnesium
Category I	74.47%	79.43%	84.39%
Category II	61.7%	72.34%	80.85%
Category III	60.99%	70.21%	63.83%

**Table 6 sensors-21-04208-t006:** Results of the soil properties prediction using a single and multi-component class strategy.

ML Method	Phosphorus		Potassium		Magnesium	
	P	PKM	K	PKM	M	PKM
SVM	60.9%	61.0%	69.5%	69.5%	73.8%	79.4%
DT	61.7%	61.7%	71.6%	72.3%	82.3%	80.9%

## Data Availability

Raw data were acquired at Faculty of Electrical Engineering, University of Ljubljana. Derived data supporting the findings of this study are available from the corresponding author on request.

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
