# Peer review of "Machine Learning Strategy for Soil Nutrients Prediction Using Spectroscopic Method"

_sensors, 2021, doi:10.3390/s21124208_

Round 1

Reviewer 1 Report

The authors here present a new machine earning strategy for soil nutrients prediction based on optical method. They found that the properties prediction accuracy is correlated with the nutrient category. The results obtained here show better performance of machine-based prediction for a dataset generated from a local farm than for a dataset of soils collected from a considerable variation of sites. The data is convincing and the discussion is based on the experiments results. However, the following should be addressed before publication of the paper.

  1. Line 108, the author declares that “drying is done naturally for a month or more”. How drying time affects experimental results?
  2. Line 365-366 “The results show an increase in prediction accuracy with decreasing categories”. The author here should explain the reason why the accuracy is increased with decreasing categories.

Author Response

Thank you for your valuable comments that helped improve the revised manuscript. 

The following changes have been made in the revised manuscript:

  1. The last keyword was removed and two added instead;
  2. Abstract, introduction and discussion were partly rewritten in order to provide a better material explanation;
  3. Section 2 and 3 titles were changed
  4. Section 2 was shortened to provide only relevant information;
  5. Figure 1 was removed as it does not provide helpful information for manuscript research purposes.
  6. The title was corrected with respect to reviewers’ recommendations
  7. Reference indexing was corrected to the new order in the revised manuscript

Answers on the comments:

1. Line 108, the author declares that “drying is done naturally for a month or more”. How drying time affects experimental results?

We agree that this sentence is confusing and can be rewritten without “or more”. Based on our experimental results, there is no difference in how dry samples are used for measurements. In making corrections for a revised manuscript, we removed this sentence.

2. Line 365-366 “The results show an increase in prediction accuracy with decreasing categories”. The author here should explain the reason why the accuracy is increased with decreasing categories.

The answer to this comment was added in the revised manuscript (Page 12, Lines 338-342).

“It was observed, that miss-classification is mainly happening for data having category properties similar to the neighbor category because of the small difference between corresponding measurements. This may explain, why the increase of the categories leads to the increase of the miss-classifications.”

Reviewer 2 Report

This article is hardly understandable. The purpose of this paper is to provide an overview of standard techniques for characterizing soil nutrient levels and highlight their advantages. So, I think it should be a review article. However, from the whole content of this paper, it seems to be a research article. The organization of this paper is quite disordered and lots of expressions are also confusing.

1) This paper didn’t describe machine learning as a tool for soil nutrients prediction in detail. The optical methods also were not well introduced. “Optical method” used in this paper should be “spectroscopy” or “spectroscopic method” more reasonable.

2) How do the authors estimate the soil nutrients according to their analysis? What are those indexes?

3) Line 98-101. It isn’t understandable. What kind of analysis could be used to estimate the influence of the nutrients level on prediction accuracy?

4) Line 153-154. The authors stated there is no more than six-level grading reported in the literature for optical methods. But more advanced levels don’t reveal.

Based on the above reasons, I could not recommend this paper for publishing in this journal.

Author Response

Thank you for your valuable comments that helped improve the revised manuscript

The following changes have been made in the revised manuscript:

  1. The last keyword was removed and two added instead;
  2. Abstract, introduction and discussion were partly rewritten in order to provide a better material explanation;
  3. Section 2 and 3 titles were changed
  4. Section 2 was shortened to provide only relevant information;
  5. Figure 1 was removed as it does not provide helpful information for manuscript research purposes.
  6. The title was corrected with respect to reviewers’ recommendations
  7. Reference indexing was corrected to the new order in the revised manuscript

Color is the revised manuscript has the following description:

  1. Insertions: blue color only
  2. Deletions: red strikethrough
  3. Moved from: green double strikethrough
  4. Moved to: green double underline

This article is hardly understandable. The purpose of this paper is to provide an overview of standard techniques for characterizing soil nutrient levels and highlight their advantages. So, I think it should be a review article. However, from the whole content of this paper, it seems to be a research article. The organization of this paper is quite disordered and lots of expressions are also confusing.

The manuscript has been reorganized with a better explanation of the material. The abstract and introduction better state the research purpose and points in the revised manuscript. The article’s organization had also been corrected to make it easier to follow the research conducted and the observation of the results.

  • This paper didn’t describe machine learning as a tool for soil nutrients prediction in detail. The optical methods also were not well introduced. “Optical method” used in this paper should be “spectroscopy” or “spectroscopic method” more reasonable.

Thank you for the sensible suggestions, which helped to improve the revised manuscript considerably. The revised manuscript was modified using the “spectroscopic method” instead of the “optical method”. Abstract, introduction and discussion were rewritten to emphasize that machine learning is just a tool used to perform research analysis.

  • How do the authors estimate the soil nutrients according to their analysis? What are those indexes?

The answer to this question was also included in the revised manuscript (Page 8, Lines 248-252).

“For example, if soil contains 5 mg of the phosphorus, 15 mg of the potassium and 35 mg of the magnesium, the soil class would be ‘2-5-11’ for Category I, ‘1-2-4’ for Category II and ‘1-1-3’ for Category III, where the first position corresponds phosphorus, the second – potassium and the third position is for magnesium.”

  • Line 98-101. It isn’t understandable. What kind of analysis could be used to estimate the influence of the nutrients level on prediction accuracy?

The referenced part was rewritten in the revised manuscript to provide a better explanation (Page 3, Lines 115-117).

“In our research, three categories having 3, 5 and 13 levels were used to validate the category influence on the prediction performance using a comparison of the machine learning results…”

  • Line 153-154. The authors stated there is no more than six-level grading reported in the literature for optical methods. But, more advanced levels don’t reveal.

The answer to this comment was included in the revised manuscript. We choose more than the 6-class category for nutrients (13-class category) to have a representative contrast with the other two categories (3-class and 5-class). Comparing categories with entirely different characteristics may provide helpful information for more advanced analysis and variations that occur in machine learning analysis. We believe that this analysis may be interesting for researchers.

Page 3, Lines 115-118:

“In our research, three categories having 3, 5 and 13 levels were used to validate the category influence on the prediction performance using a comparison of the machine learning results, where 13 levels were selected to have a representative contrast to other two.”

Page 13, Lines 383-386:

“We believe that several categories may be needed not only to characterize the nutrients in the soil, but also to characterize measuring technique sensitivity and selectiveness. In this research 13-class category characterization was used based on our project requirements.”

Reviewer 3 Report

Thank you for have the time to publish such an important material.

However, I missed the scientific writing style in this paper. In the current version it looks more like a research company report than an actual paper to be published in a journal.

My recommendations in this regard are that the authors should structure the paper with hypothesis, goals and objectives at the same time that the conclusions are linked to them. Furthermore, these conclusions should be based on the results (which are still not super clear to me).

After a major revision on the form of the paper, it may be considered for accpetance.

Thank you!

Author Response

Thank you for your valuable comments that helped improve the revised manuscript

The following changes have been made in the revised manuscript:

  1. The last keyword was removed and two added instead;
  2. Abstract, introduction and discussion were partly rewritten in order to provide a better material explanation;
  3. Section 2 and 3 titles were changed
  4. Section 2 was shortened to provide only relevant information;
  5. Figure 1 was removed as it does not provide helpful information for manuscript research purposes.
  6. The title was corrected with respect to reviewers’ recommendations
  7. Reference indexing was corrected to the new order in the revised manuscript
  • My recommendations in this regard are that the authors should structure the paper with hypothesis, goals and objectives at the same time that the conclusions are linked to them. Furthermore, these conclusions should be based on the results (which are still not super clear to me).

We thank you for your valuable suggestions. The revised manuscript was significantly improved. The abstract, introduction and discussion were rewritten to provide a better explanation. The main research points were made more pointed in the revised version. Some sections were reorganized, irrelevant figures and text were removed. We hope that the revised manuscript will conform to the format of a research journal and is suitable for publication.

Reviewer 4 Report

The study is interesting, and the findings can be useful as a general guidance for researchers intending to use optical tools to estimate soil nutrients. However, several flaws must be addressed.

Line 8: The abstract is confusing. It does not mention what techniques are evaluated in this study. Very important information to be placed in the abstract. Is it a review?

Line 48: What region?

Line 97: What is the hypothesis of this study?

Line 103: It is necessary to add more details of how the search was conducted? What databases were used? What search terms? Keywords?

Line 262: Why these 3 elements were selected? Why only phosphorus, potassium, and magnesium?

Line 348: Is it a review? If yes, you need to provide much more details in the material and methods of how it was done. Also, the first time that this paper was categorized as a review was in the conclusion section. It must me mentioned across the manuscript and perhaps in the title. In my opinion, I would categorize this manuscript as an overview or compilation of studies.  

Line 349: Systematic review? What type of review?  

Author Response

Thank you for your valuable comments that helped improve the revised manuscript

The following changes have been made in the revised manuscript:

  1. The last keyword was removed and two added instead;
  2. Abstract, introduction and discussion were partly rewritten in order to provide a better material explanation;
  3. Section 2 and 3 titles were changed
  4. Section 2 was shortened to provide only relevant information;
  5. Figure 1 was removed as it does not provide helpful information for manuscript research purposes.
  6. The title was corrected with respect to reviewers’ recommendations
  7. Reference indexing was corrected to the new order in the revised manuscript
  • Line 8: The abstract is confusing. It does not mention what techniques are evaluated in this study. Very important information to be placed in the abstract. Is it a review?

The abstract was rewritten in the revised manuscript to provide more information about the techniques used. This paper aimed to provide a research analysis of the common factors that influence machine learning of spectroscopy measurements. We hope that the revised manuscript will provide a better description of this research purpose and points. 

  • Line 48: What region?

We appologize for the incorrect expression. The required correction was made in the revised manuscript (Page 2, Lines 45-46).

“Nevertheless, UV-VIS-NIR range contains helpful information about organic and inorganic materials in the soil [8].”

  • Line 97: What is the hypothesis of this study?

The abstract and introduction were rewritten in order to provide better research points (hypothesis). They are mentioned in the introduction and shortly summarized below:

1.“The first research point is to determine an appropriate machine learning technique for the most accurate soil properties prediction …” (Page 2, Lines 59-60).

2.“The next research point is to investigate the Principal Components (PCs) influence on the overall performance …” (Page 2, Lines 81-82).

3.“Another research point is to determine an appropriate category for machine learning without significant accuracy loss …” (Page 2, Lines 90-91).

4.“Finally, the influence of the soil location on the prediction accuracy was investigated.…” (Page 3, Line 1120).

  • Line 103: It is necessary to add more details of how the search was conducted? What databases were used? What search terms? Keywords?

The revised manuscript includes a better explanation of the database and search terms.  The manuscript was also improved technically for a better explanation about how the search was conducted. The ‘artificial intelligence’ was removed from the keywords as it was not discussed in detail in the manuscript. The research dataset is now described in a separate paragraph “A. Research dataset” (Section 3). 

Keywords were also corrected to fit the content in the revised manuscript better.

“Keywords: machine learning; nutrients prediction; soil spectra; soil analysis; soil category; precision farming;”

  • Line 262: Why these 3 elements were selected? Why only phosphorus, potassium, and magnesium?

The answer to this question was added in the revised manuscript (Page 3, Lines 133-136 and Page 13, Lines 387-391).

“Due to the project requirements, the only three nutrients were selected for analysis: Phosphorus (P), Potassium (K) and Magnesium (M) as they are the most common for local agriculture in vineyard and orchards.”

“The presented study was performed only for phosphorus, potassium and magnesium. Nevertheless, similar analysis can be repeated for other soil components such as nitrogen and organic mass. We believe that presented results can clarify the understanding of the most common parameters that affect the machine learning performance and help to select the optimal strategy for a particular research task in soil study.”

  • Line 348: Is it a review? If yes, you need to provide much more details in the material and methods of how it was done. Also, the first time that this paper was categorized as a review was in the conclusion section. It must me mentioned across the manuscript and perhaps in the title. In my opinion, I would categorize this manuscript as an overview or compilation of studies.  

We are sorry that the article looks like a review because it was not intended to be. After performing corrections, the revised manuscript highlights that this is a research article describing the results of our comparative analysis on strategy development in predicting soil properties using machine learning and spectroscopic methods. 

To emphasize that this is a research paper, the revised manuscript introduction and abstract were rewritten. Section 2 title was changed to “Advantages and Disadvantages of Machine Learning”.

  • Line 349: Systematic review? What type of review?  

As was answered in the previous question, the paper was not intended to be a review manuscript.

Round 2

Reviewer 2 Report

The manuscript has been well revised and could be recommended for publication.

Author Response

The revised manuscript language and style were corrected. It was proofread by an academically educated person whose native language is English and spell check was verified with the Grammarly writing assistant software.

Reviewer 4 Report

This study is still very confusing. The hypothesis is not clearly stated in the abstract nor introduction sections. We would like to learn what are the expected outcomes of the study therefore the reason to have this research done. The materials and methods are not precisely describing how the techniques were used and applied.

Author Response

We apologise that part of the revised manuscript is still unclear.  We hope that the revised manuscript provides a more straightforward material explanation and answers the questions. The following changes were made:

  • The abstract and introduction were improved to highlight the research hypothesis.
  • Unimportant material was removed.
  • The description of the dataset was improved.
  • A separate paragraph (B. Spectroscopic data acquisition) was created to describe how the measuring procedure was performed.
  • Section 3 was reorganised into two Sections, “Materials and Methods” and “Results”, to improve the paper organization.
  • Section “Discussion and Conclusion” was divided into two sections, 5 and 6, to improve the paper organization.
  • The article was proofread by an academically educated person whose native language is English and spell check was verified with the Grammarly writing assistant software.
  • The most critical changes are highlighted with yellow in the revised manuscript.
  • Figure 2 was revised.

Round 3

Reviewer 4 Report

Good efforts to improve it, but as another reviewer have mentioned, this manuscript is compiled in a very confusing way.